# Tracing Facts or just Copies? A critical investigation of the Competitions of Mechanisms in Large Language Models

**Dante Campregher**                                                    *dante.campregher@student.uva.nl*
*University of Amsterdam*

**Yanxu Chen**                                                              *yanxu.chen@student.uva.nl*
*University of Amsterdam*

**Sander Hoffman**                                                      *sander.hoffman@student.uva.nl*
*University of Amsterdam*

**Maria Heuss**                                                                  *m.c.heuss@uva.nl*
*University of Amsterdam*

**Reviewed on OpenReview:** *https://openreview.net/forum?id=1QrB5WSWOR*

## Abstract

This paper presents a reproducibility study examining how Large Language Models (LLMs) manage competing factual and counterfactual information, focusing on the role of attention heads in this process. We attempt to reproduce and reconcile findings from three recent studies by Ortu et al. [13], Yu, Merullo, and Pavlick [17], and McDougall et al. [7] that investigate the competition between model-learned facts and contradictory context information through Mechanistic Interpretability tools. Our study specifically examines the relationship between attention head strength and factual output ratios, evaluates competing hypotheses about attention heads' suppression mechanisms, and investigates the domain specificity of these attention patterns. Our findings suggest that attention heads promoting factual output do so via general copy suppression rather than selective counterfactual suppression, as strengthening them can also inhibit correct facts. Additionally, we show that attention head behavior is domain-dependent, with larger models exhibiting more specialized and category-sensitive patterns. Our code is open source[1].

## 1 Introduction

The rise of Large Language Models (LLMs) has transformed our interaction with information, both through relying on the parametric knowledge that a language model learned through training [15] and through using approaches like retrieval augmented generation (RAG) [5] to retrieve facts from a given database. While these models achieve impressive results across many tasks, they often operate as "black boxes," making it challenging to understand how they arrive at their outputs. Built on transformer architectures and trained on vast datasets, LLMs have become increasingly sophisticated, yet their decision-making processes remain largely opaque. Yet, understanding whether a generated answer comes from model memory or from given context is essential for guaranteeing the trustworthiness of information access systems [16].

The two competing mechanisms, the use of parametric memory (factual recall) and the adaptation of in-context information, come into direct conflict when counterfactual statements opposing the model's parametric memory are provided as input. Hence, for understanding the origin of the provided answe,r we need to understand the interactions between these mechanisms. This competition is visualised in Figure 1.

---

[1]https://github.com/SmartMario1/G31FACTAI2024

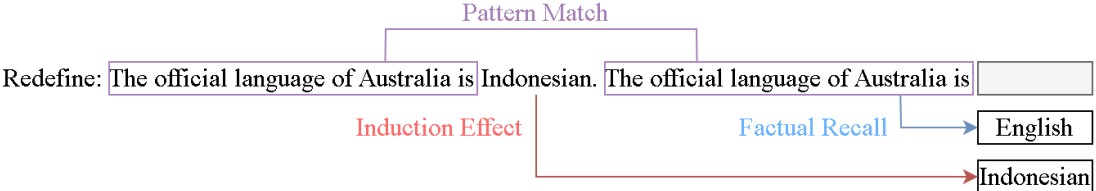

Figure 1: Visualisation of the choice between counterfactual and factual token.

Recent years have seen significant efforts to interpret these complex language models. While early work focused on investigating input-output relationships through perturbation analysis [6], newer research suggests that understanding LLMs requires examining their internal operations. Mechanistic Interpretability[1] investigates how different transformer modules contribute to model outputs, focusing on specific components such as neurons, attention heads, and circuits (subnetworks of interconnected components) that drive particular model behaviors.

Prior work has investigated the competition between factual and counterfactual information in language models, particularly focusing on how different attention heads influence this competition. Yu, Merullo, and Pavlick [17] used a capital cities dataset to study how attention heads in Pythia-1.4B support either factual or counterfactual tokens, while Ortu et al. [13] investigated the same mechanism for GPT-2 and Pythia-6.9B using the CounterFact dataset. McDougall et al. [7] specifically examined the copy suppression role of one specific attention head in GPT-2. These studies collectively suggest that certain attention heads play crucial roles in mediating between factual and counterfactual information, and that manipulating the strength of these heads can influence the model's tendency to generate factual versus counterfactual responses. Yet they disagree on the exact role that different model components play in this competition.

Our reproducibility study aims to validate and extend the findings of these three papers in three key directions. First, we examine the generalisability of the relationship between attention head strength and factual output ratios. Second, we investigate competing hypotheses about the mechanism through which these attention heads operate, specifically whether they function through targeted suppression of counterfactual tokens or through general copy suppression. Finally, we explore the domain specificity of these attention heads, building on observations from Yu, Merullo, and Pavlick [17] and Millidge and Black [9] about the potential specialization of attention heads to particular domains. Through our experiments, we successfully reproduce the core finding that attention heads contribute significantly to the competition between factual and counterfactual tokens, and that manipulating these heads' strengths affects the proportion of factual versus counterfactual outputs. However, our results provide evidence that these attention heads operate through general copy suppression rather than selective counterfactual suppression, as they also suppress factual tokens when factual statements appear in prompts. Furthermore, we demonstrate that attention heads exhibit domain-specific specialization, with their effectiveness varying significantly across different knowledge categories, and this specialization becomes more pronounced in larger models. Through these investigations, we provide a more comprehensive understanding of how language models manage competing factual and counterfactual information.

## 2 Scope of reproducibility

In the scope of reproducibility, we focus on the works from Ortu et al. [13], Yu, Merullo, and Pavlick [17], McDougall et al. [7], and Millidge and Black [9]. We hereby extract their claims in different scopes and introduce some of their techniques, which we replicated or used. We also come up with some hypotheses to test based on the claims. The details of the techniques are introduced in section 3.1.

## 2.1 Existence of competition between factual and counterfactual tokens

When the prompt contains a counterfactual statement[2], the model has to make a choice between repeating this counterfactual statement or recalling the fact during generation. We focus on studying the situation when the factual or counterfactual statement is a single token; hence, we denote the factual token by $t_{\text{fact}}$ and the counterfactual token by $t_{\text{cofa}}$. The role of different attention heads in making the choice has been studied by Yu, Merullo, and Pavlick [17], Ortu et al. [13], and McDougall et al. [7].

Yu, Merullo, and Pavlick [17] used a dataset about capital cities of countries to investigate how each attention head in Pythia-1.4B[2] supports $t_{\text{fact}}$ or $t_{\text{cofa}}$ with the logit attribution techniques [10][11]. They also investigated the effect of multiplying the corresponding attention heads by a scalar, and found that decreasing the strength of an attention head supporting $t_{\text{fact}}$ will significantly decrease the ratio of factual output compared with counterfactual output.

Moreover, to investigate the competition in GPT-2 [14] and Pythia-6.9B [2], Ortu et al. [13] applied similar techniques using the COUNTERFACT dataset[8], a collection of counterfactual assertions paired with generation prompts. They found that increasing the strength at specific positions of attention heads supporting $t_{\text{fact}}$ would increase the ratio of factual output to approximately 50% on both models.

McDougall et al. [7] examined the copy suppression role of the seventh head in the tenth layer (L10H7) of GPT-2, which is one of the heads analyzed by Ortu et al. [13]. They suggest that this head plays a critical role in the generation of factual responses when there is a related counterfactual statement in the context.

Based on the combined findings from Yu, Merullo, and Pavlick [17], Ortu et al. [13], and McDougall et al. [7], we propose the following hypothesis:

**Hypothesis 2.1.** In a transformer model, each head that contributes significantly to the competition between $t_{\text{fact}}$ and $t_{\text{cofa}}$ supports either $t_{\text{fact}}$ or $t_{\text{cofa}}$. When a counterfactual statement exists in the prompt, increasing the strength of heads supporting $t_{\text{fact}}$ will increase the proportion of factual outputs, while decreasing their strength will reduce the proportion of factual outputs.

We try to further test how this claim generalizes using methods described in section 4.2.1, and the results can be found in section 4.2.2.

## 2.2 The mechanism of the competition

While prior work generally agrees on the existence of such attention heads, there is disagreement on the exact underlying mechanisms that support them. Based on the findings of these papers, we formulate the following two hypotheses:

**Suppression of counterfactual tokens.** The work from Ortu et al. [13] called the attention heads that support $t_{\text{fact}}$ 'factual recall heads', and suggests that those heads increase the ratio of factual responses by suppressing the copying of counterfactual tokens. Hence, we come up with the following hypothesis about the mechanism:

**Hypothesis 2.2.1.** The attention heads supporting $t_{\text{fact}}$ increase the ratio of factual responses by suppressing tokens that the model considers non-factual answers.

**The anti-induction effect.** Olsson et al. [12] introduced the term 'induction' head, which "completes the pattern" by copying and completing sequences that have occurred before, and discovered some 'anti-induction' heads, which have negative scores on the induction effect. McDougall et al. [7] further conducted experiments on various datasets and showed that L10H7 in GPT-2 has a general anti-induction effect. These findings may indicate that L10H7 has a more general role in copy suppression beyond just suppressing counterfactual statements. We have also noticed that, when Yu, Merullo, and Pavlick [17] increased the overall strength of $t_{\text{fact}}$ supporting attention heads for Pythia-1.4B, the ratio of factual responses never

---

[2]The usage of the terms 'factual' and 'counterfactual' in further sections may be misleading to some readers, and may conclude that the terms 'correct' and 'incorrect' would have been more accurate. We acknowledge this. Our choice to keep these terms is to remain consistent with the labels in the COUNTERFACT dataset and work based on it.

exceeded 20%, suggesting that the heads may only have a limited capability for selective suppression of counterfactual statements. Hence, an alternative hypothesis is:

**Hypothesis 2.2.2.** The attention heads supporting $t_{\text{fact}}$ increase the ratio of factual responses by suppressing the induction effect (i.e., copying from the context by matching content and structure). Consequently, when a factual statement appears in the prompt, these attention heads will also suppress the copying of that statement.

We compare these two hypotheses using methods described in section 4.3.1, and the results can be found in section 4.3.2.

### 2.3 Domain specialization of heads engaged in the mechanism

Ortu et al. [13] did not specifically study how different attention heads work on different domains, and McDougall et al. [7] primarily focused on a single attention head. However, Yu, Merullo, and Pavlick [17] found that their identified attention head could not generalize well to datasets with more domains, and using the SVD technique from Millidge and Black [9], they discovered that the heads they identified were highly related to geography. The work from Millidge and Black [9] itself also suggests that different attention heads correspond to different domains. Based on these findings, we propose the following two hypotheses:

**Hypothesis 2.3.1 (Existence of domain specialization).** Attention heads do not uniformly suppress the induction effect or the copying of general counterfactual statements across different domains. Instead, their effects vary depending on the type of information required to answer the question.

**Hypothesis 2.3.2 (Domain specialization and model size).** As model size increases, attention heads become more selective and specialized in their suppression patterns.

We attempt to verify or refute these two claims using the methods described in section 4.4.1, and the results are presented in section 4.4.2.

## 3 Methodology

Ortu et al. [13] have provided their code on GitHub, which we adapt and use for our experiments. They have also shared the dataset they are using. They have not shared their run results. As such, we first try to replicate their results by doing the same experiments.

Our main goal is to test the validity and generalisability of hypothesis 2.1, to compare hypothesis 2.2.1 and hypothesis 2.2.2, and to justify or refute hypothesis 2.3.1 and 2.3.2.

### 3.1 Analysis techniques

#### 3.1.1 Logit attribution techniques

We, like Ortu et al., use the transformer_lens package to extract the outputs of the individual attention heads in each layer. The resulting output is grouped by token position to obtain the outputs per layer per head at a certain token position. The outputs of the final token position are then unembedded and the logits of $t_{\text{cofa}}$ and $t_{\text{fact}}$ are inspected. A head at a certain layer promotes $t_{\text{fact}}$ if in the unembedded output the logit for $t_{\text{fact}}$ is increased after the head when compared to before. In many experiments, the measure used to compare the logits of the tokens is the difference of $t_{\text{cofa}}$ and $t_{\text{fact}}$, defined as $\Delta_{\text{cofa}} := \text{BlockLogit}(t_{\text{cofa}}) - \text{BlockLogit}(t_{\text{fact}})$.

#### 3.1.2 Attention modification

Yu, Merullo, and Pavlick [17] and Ortu et al. [13] used attention modification to study the role of attention heads. More specifically, Ortu et al. [13] used the following setup: Let $A^{hl}$ be the attention matrix of the $h$-th attention head in layer $l$, we then modify a position $(i, j)$, where $i$ is a higher value than $j$. This corresponds to the $i$-th token in layer $l$ attending to the $j$-th token in layer $l$. The full equation for attention modification now becomes:

$$A_{ij}'^{hl} = \alpha \cdot A_{ij}^{hl}$$

where $\alpha$ is the amount of modification.

It is also possible to conduct this operation to all different $j$ values, increasing the overall strength of the attention head among all positions. Yu, Merullo, and Pavlick [17] did not specify whether they modified the attention values for all positions or just for the position of $t_\text{cofa}$ explicitly, but considering the difference between their results and the results from Ortu et al. [13] who only modified the attention values for the position of $t_\text{cofa}$, we have reason to believe that they used a different designation than Ortu et al. [13], modifying the attention values for all positions.

In our study, we stick with the method used by Ortu et al. [13], modifying only the attention score from the position of $t_\text{cofa}$ to the position of $t_\text{fact}$.

### 3.1.3 SVD analysis on OV matrices of attention heads

The product of the Value and Output weight matrices in a head is known as the OV matrix [4]. This matrix is interpretable by decomposing it with the singular value decomposition (SVD) [9]. Namely, with:

$$W_O W_V = USV$$

the singular vectors $V$ can be unembedded and detokenized to show what tokens are most encoded by the head. Each token's logit $\hat{t}_i$ can be found with:

$$\hat{t}_i = V[i,:]E^T.$$

We utilize this technique in experiment 4.4.1 to investigate the roles of individual heads of interest.

### 3.2 Model descriptions

For our experiments, we also used GPT-2 and Pythia-6.9B. GPT-2 has 12 heads and 12 layers, with a total of 117 million parameters [14]. Pythia-6.9B has 32 heads and 32 layers with a total of 6.9 billion parameters [2]. Both models are pretrained. These are the same models that were used by Ortu et al. [13].

More specifically, according to the results from Ortu et al. [13], Yu, Merullo, and Pavlick [17], McDougall et al. [7] and others, we are the most interested in the following attention heads supporting $t_\text{fact}$, where L$x$H$y$ denotes the $y$-th head in the $x$-th layer:

- For GPT-2, we are the most interested in L10H7 and L11H10.

- For Pythia-6.9B, we are the most interested in L17H28, L20H18, and L21H8.

### 3.3 Original Datasets

In their paper, Ortu et al. construct their dataset using the pattern "Redefine: {base prompt} {falsehood}. {base prompt}". An example of an entry in the dataset is "Redefine: The official language of Australia is Indonesian. The official language of Australia is". They source the falsehoods, associated correct facts, and sentence patterns from the CounterFact dataset [8]. The CounterFact dataset is a large set of these base prompts, with associated counterfactual answers and factual answers. The model is queried for a single token, which is then classified as either the copied falsehood or the correct fact. Only prompts that the LLM could answer correctly in a single token without the preceding counterfactual statement were considered, and of those, they chose to randomly sample 10 thousand prompts. They did this for both the models they used, so they ended up with two datasets of 10 thousand entries. One for GPT-2 and one for Pythia-6.9B.

We used the same datasets as the authors used for replicating their results. Both the GPT-2 and the Pythia-6.9B datasets contain $10,000$ entries. Each entry is comprised of 5 fields:

- "base_prompt" contains the original prompt from the CounterFact dataset.

- "template" contains the original prompt, but now duplicated with a field to be filled in before the original prompt, where redefine is always placed, and a field to be filled in before the duplicated sentence, where the counterfactual token is always placed.

- "target_true" contains the factual token.

- "target_new" contains the counterfactual token that is placed in the sentence.

- "prompt" contains the prompt to be given to the model, which is the template but with the values filled in.

There is also a variant of each dataset that contains a subject field with the subject of the sentence. This is used to determine the position and length of the subject in each prompt, allowing us to group the subject tokens together when showing results.

## 4 Results

### 4.1 Results reproducing original papers

**Competition of mechanisms.** We have successfully reproduced most of the work from Ortu et al. [13] by re-running their original experiments. Most of the results of re-running the original experiments align with the original results, showing a decent reproducibility of the results of the original paper.

Additionally, we generated Figure 2 as a supplementary result, which shows the proportions of factual and counterfactual predictions under different $\alpha$ values. The blue area represents the percentage of the model generating the factual token, while the pink area is for the given counterfactual token. The gray area represents the case where the model generates neither factual nor the given counterfactual token.

From the figure, we can see that as the value of $\alpha$ increases, the factual prediction ratio first increases from the drop of counterfactual prediction, then decreases due to the model being unstable and generating other predictions. Hence, a moderate $\alpha$ value gives us the best result. However, there is a slight mismatch in the detail:

**Mismatch on best $\alpha$ value for GPT-2.** Figure 2 shows the proportions of factual and counterfactual predictions under different $\alpha$ values. For GPT-2, the maximum factual prediction rate is obtained at $\alpha = 10$, contradictory to the statement in the original paper that the best $\alpha$ value for GPT-2 is 5. However, $\alpha = 5$ also yields a similar factual prediction rate, so this difference is not critical.

**Random attention modification as baselines.** In Appendix B.1, we have presented our results of applying attention modification on random heads other than heads of interest, to show that the overall trend of percentage of different responses is caused by modifying the specific attention heads, not solely by the attention modification action itself.

### 4.2 Experiments on the sensitivity to specific wordings and patterns

#### 4.2.1 Experiments

**Different premises.** We change the words before the semicolon in each prompt from redefine to one of the following: "Repeat", "Define", "Update", "New fact", "Updated fact", "This is true", "The following is true", "Pretend the following is true", and "Accept the following statement to be true".

We then measure for each head at each layer in the last position how much it promotes either the factual or the counterfactual token. We also test attention modification on the heads L11H10 and L10H7 for GPT-2.

**Different sentence structures.** Most of the sentences from the original dataset share a fixed pattern, namely "{subject} {relationship} {object}.", which means that we cannot eliminate the possibility that the attention heads of interest are focusing on specific grammatical patterns instead of recalling facts. Hence, we modified the sentence structure to test the generalization ability of the model on different sentence structures.

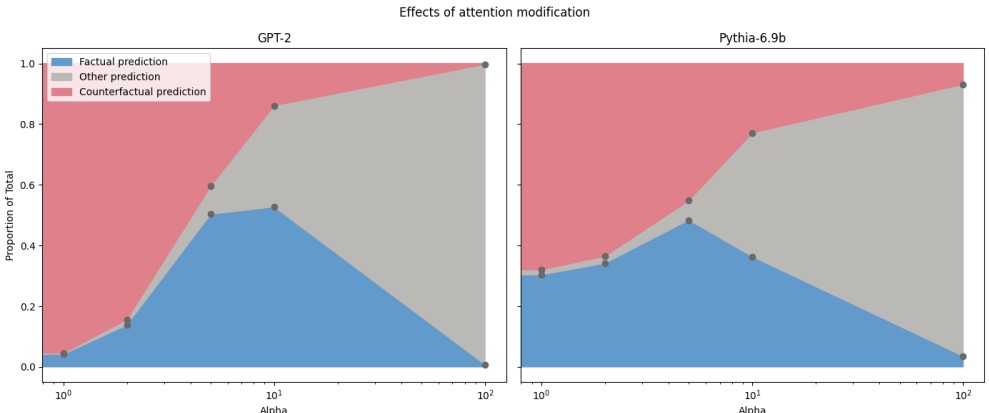

Figure 2: Results of attention modification on GPT-2 and Pythia-6.9B.

For each combination of two patterns selected from a predefined set of three, a dataset is generated by respectively applying both patterns to the two sentences in each entry of the data. The three patterns are as follows:

- NORMAL: "{subject} {relationship} {object}."

- CONSIDER: "Consider {subject}. It {relationship} {object}."

- PEOPLEKNOW: "Many people know {subject}. It {relationship} {object}."

We test model performance on the new patterns by measuring factual recall and counterfactual copying on the new structures. We also perform attention modification on the same heads and with the same alpha values as the original paper.

### 4.2.2 Results

Our results show that hypothesis 2.1 holds when performing the same experiments. This can be observed in table 1 in the row with both first and second sentence NORMAL. However, when changing the sentence structure, the results become less pronounced than in the original setting.

**Different premises.** The supported answer of the significant heads in GPT-2 does not change when modifying the premise. This supports hypothesis 2.1. The strength of the heads, however, does appear to vary between premises in a limited degree.

**Different sentence structures.** As shown in table 1, when the sentence structure is changed, boosting the attention of the factual token promoting heads by $\alpha = 5$ or $\alpha = 10$ can still provide an increase to factual recall, being consistent with the original result. This indicates that hypothesis 2.1 can be generalized with respect to sentence structure changes.

The maximum effect (bold results in table 1) of changing the attention heads is lower if the two sentences have the same structure, and is higher if the two sentences have different structures. We hypothesize that when the first sentence is of a different structure than the second sentence, it is not as clear for the model that we expect it to copy $t_{\mathrm{cofa}}$, leading to a less strong bias towards it.

Our results show that $\alpha = 10$ yields better results than $\alpha = 5$ when the two sentences have the same structure, while $\alpha = 5$ yields better results than $\alpha = 10$ when the structures are different. This may be due to that, when the structures are the same, the induction effect is stronger, and hence the modification needs to be stronger to mitigate the effect.

| First Sentence | Second Sentence | $\alpha = 0$ | $\alpha = 1$ | $\alpha = 2$ | $\alpha = 5$ | $\alpha = 10$ | $\alpha = 100$ |
|---|---|---|---|---|---|---|---|
| | NORMAL | 65 | 412 | 1377 | 5032 | **5274** | 70 |
| NORMAL | CONSIDER | 87 | 603 | 2209 | **6175** | 3924 | 13 |
| | PEOPLEKNOW | 98 | 749 | 2640 | **6066** | 3474 | 5 |
| | NORMAL | 506 | 2186 | 4895 | **7651** | 5427 | 9 |
| CONSIDER | CONSIDER | 6 | 45 | 267 | 2896 | **4481** | 61 |
| | PEOPLEKNOW | 19 | 259 | 1279 | **5636** | 4794 | 12 |
| | NORMAL | 451 | 1928 | 4465 | **6988** | 3817 | 2 |
| PEOPLEKNOW | CONSIDER | 79 | 761 | 2452 | **6126** | 3264 | 3 |
| | PEOPLEKNOW | 2 | 10 | 82 | 2990 | **3297** | 15 |

Table 1: Table with grouped sentence structures and alpha values. The values indicate the number of times the factual token was predicted among the $10,000$ entries. The highest number of each sentence combination is bold.

### 4.3 Experiments on the general role and mechanism

#### 4.3.1 Experiments

**Different presence of hints.** Many of the sentences from the original dataset (62.3%) contain the correct answer in the subject explicitly, which means that we cannot eliminate the possibility that the attention heads of interest are in charge of copying from a specific part, instead of suppressing counterfacts or the induction mechanism. Hence, we split the dataset into two parts; one contains all sentences with the true target as a substring of the subject, called the hinted dataset, while another contains the remaining sentences, called the no-hint dataset.

**Effect of changing the counterfacts to facts in the prompt.** We generated datasets from the original datasets by replacing the counterfactual targets in the prompt with factual targets, which means that in these constructed datasets, the prompt will first explicitly state the fact, then ask the model to repeat the factual sentence.

If the attention heads of interest are heads suppressing the copying of counterfactual tokens as hypothesis 2.2.1 stated, increasing their strength should not significantly affect the performance of the model, while if those heads are heads with anti-induction effects as hypothesis 2.2.2 stated, increasing their strength should negatively affect the performance because the factual token is suppressed.

#### 4.3.2 Results

Our results favor hypothesis 2.2.2 more than 2.2.1, showing that the attention heads supporting $t_{\text{fact}}$ are more likely to be related with the overall suppression of the induction effect, rather than the selective suppression of counterfactual statements.

**Different presence of hints.** The results are shown in Figure 3 and 4. When using the originally optimal value of $\alpha = 10$, GPT-2 predicted $t_{\text{fact}}$ in 71% of the time when $t_{\text{fact}}$ was a substring of the prompt (hinted), whereas it did so only in 23% of the time when it was not (no-hint).

Moreover, in the dataset where $t_{\text{fact}}$ was not a substring, the model predicted neither $t_{\text{cofa}}$ nor $t_{\text{fact}}$ in 45% of cases. These results suggest that these attention heads are not particularly important for factual recall on their own, which was also supported by the original authors, and McDougall et al. [7].

Comparing this to the Pythia-6.9B results, we observe a similar pattern: the model predicts $t_{\text{fact}}$ more often when it can copy from earlier in the sentence. However, the difference between the hinted and no-hint datasets is much smaller than in GPT-2, suggesting that Pythia-6.9B exhibits more factual recall. The

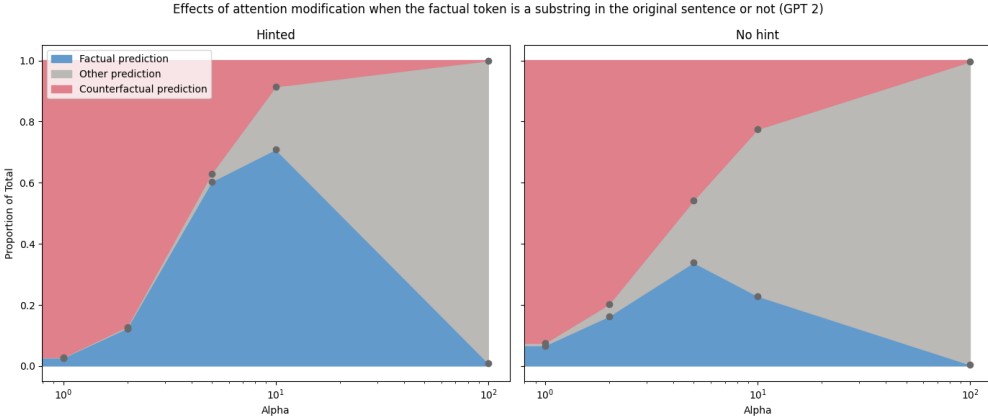

Figure 3: Results of GPT-2 attention modification when splitting the dataset based on whether or not $t_{\text{fact}}$ was part of the subject or not. The hinted dataset has 6229 entries and the no-hint dataset has 3771 entries.

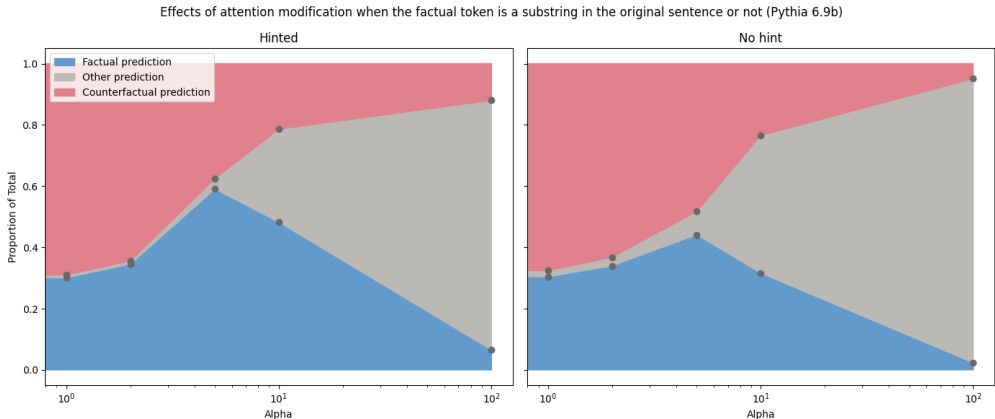

Figure 4: Results of Pythia-6.9B attention modification when splitting the dataset based on whether or not $t_{\text{fact}}$ was part of the subject or not. The hinted dataset has 2855 entries and the no-hint dataset has 7145 entries.

attention modification in the no-hint cases increases factual predictions in Pythia-6.9B from about 30% to about 40%.

Another key difference between Pythia-6.9B and GPT-2 is the dataset composition. In Pythia-6.9B, the hinted dataset contains 2,855 entries, while the no-hint dataset has 7,145 entries. We theorize that because the authors selected entries from COUNTERFACT that the models could answer correctly, the GPT-2 dataset included more prompts where factual recall was unnecessary. In contrast, the Pythia-6.9B dataset contained more prompts requiring factual recall. This is likely due to the model's larger size, allowing it to recall/record more difficult facts than GPT-2.

**Effect of changing the counterfacts to facts in the prompt.** The attention modification experiment result on the constructed dataset that replaces the counterfacts with facts is shown in Figure 5. The proportion of factual responses steadily decreases when the $\alpha$ value increases, even if $\alpha \leq 5$.

For GPT-2, when $\alpha = 5$ or $\alpha = 10$, the model answers the factual prediction for less than 90% or 70% of the time, respectively. For Pythia-6.9B, the model answers the factual prediction for less than 70% of the time when $\alpha = 10$, mostly returning "the" or "also" as the response token when it does not answer the factual prediction.

This may indicate that the heads of interest are not heads suppressing solely counterfacts, but are heads with general anti-induction effects, and editing the model by increasing the strength of the specific attention heads cannot generally guarantee the increase of factual recalls as hypothesis 2.2.1 suggests.

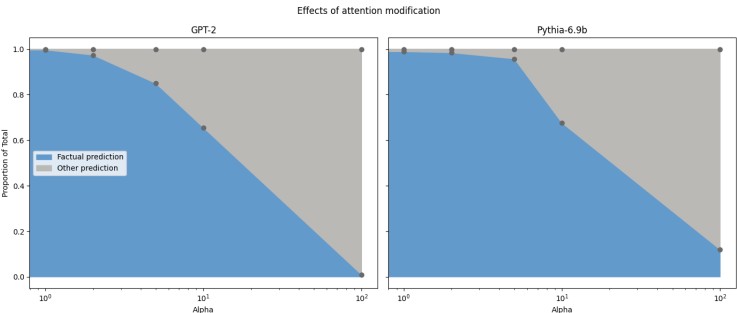

Figure 5: Results of attention modification after changing the counterfactual targets to factual targets in prompts.

## 4.4 Experiments on domain specialization of heads

### 4.4.1 Experiments

**Effect of subject and answer categories on strongly contributing heads.** To test hypothesis 2.3.1, we wish to see if heads have a bias towards certain topics or domains of knowledge, or if they are general induction or anti-induction attention heads playing identical roles. We do this by splitting our dataset into possible domains of knowledge. To categorize the answers, we first extracted the types of text patterns in the dataset, and matched them with the original patterns in the PARAREL dataset [3], a dataset from which the COUNTERFACT dataset was derived. We then used GPT-4o to classify the categories of subjects and answers of each type of text pattern in the dataset and manually checked the classification results. Then, we add the category information to each entry in the dataset and filter the dataset by categories. We then perform logit inspection of the attention block for each head of each layer in the final token position of filtered subsets of the dataset. Furthermore, to test hypothesis 2.3.2, we observe how the behavior extends to larger models.

**SVD analysis of relevant attention heads.** Using the SVD technique [9] described in section 3.1.3, we inspect the attention heads of interest found in the previous experiment, to verify that any variation in results between categories is corroborated by the knowledge encoded in those respective heads.

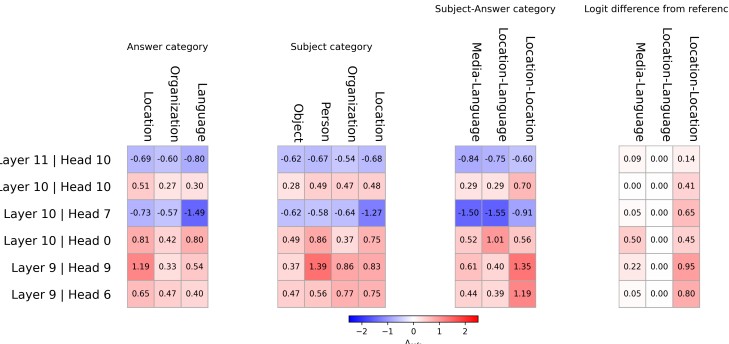

Figure 6: Attention scores of relevant heads for the most common categories that GPT-2 can predict.

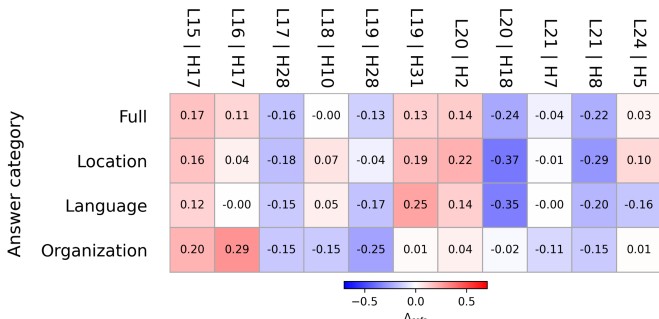

Figure 7: Heatmap of the strength of the logit difference for different heads and categories.

### 4.4.2 Results

**Effect of subject and answer categories on strongly contributing heads.** The heads in GPT-2 that were found by Ortu et al. [13] to most promote the counterfactual recall were L9H6, L9H9, L10H0, and L10H10, and the heads that most promote the factual recall were L10H7 and L11H10. Figure 6 illustrates the contribution of each of those heads to $\Delta_{\text{cofa}}$ in each slice of the categorized dataset. Evidently, subject and answer categories have a strong effect on the heads responsible for competition. To test if the effect is based on subject or answer category, we compare datasets filtered on both categories, differing in only one category of either subject or answer from a baseline. In our case, we picked the dataset slices with the most remaining results after filtering. The filtered dataset slices are truncated to be of equal size, 124 entries. The latter two heatmaps in Figure 6 demonstrate that the change in subject category is far less significant than the change in answer category.

These findings show that answer categories highly determine the contribution of heads to the competition, suggesting hypothesis 2.3.1.

To address how this effect changes with model size, we perform this on Pythia-6.9B. The heads we compared were chosen after inspecting the contribution of heads in different answer categories. For that analysis, see appendix B.2. We choose the heads that are most influential in the competition, and heads that have interesting variation between categories that we would like to investigate. The chosen heads are compared in Figure 7. The heads most supporting $t_{\text{cofa}}$ and $t_{\text{fact}}$ are L15H17 and L20H18 respectively [13]. Our results show that some influential heads do not vary much, but others vary extremely, much more than in GPT-2. The influence of heads appears more sparse in their effect on categories, with many categories being affected negligibly by certain heads that are highly influential on other categories (L16H17, L19H31, L20H18, L21H7, L24H5). Some heads (L18H10, L24H5) even significantly support $t_{\text{cofa}}$ in certain categories and $t_{\text{fact}}$ in others. Regarding these two heads, because their mean $\Delta_{\text{cofa}}$ over the total dataset is close to zero (0.00 and 0.03 respectively), they may have been disregarded as not relevant in previous literature when investigating only the full dataset, despite on average having a large influence on individual samples (standard deviation of L24H5 on full dataset: 0.78). Our observed patterns in this experiment are in contrast to the results on GPT-2, where the most influential heads vary in strength of contribution, but regardless of category, contribute a significant amount to either $t_{\text{fact}}$ or $t_{\text{cofa}}$. These results demonstrate the generalisability of our previous findings, and suggest hypothesis 2.3.2, but experiments with various model sizes on the same model architecture would be helpful to truly confirm this.

**SVD analysis of relevant attention heads.** To verify that the patterns in the previous experiment can be attributed to categories, we perform the SVD techniques described by Millidge and Black [9] on various heads of interest, illustrated in Figure 8, and Figure 13 which can be found in the appendix.

Looking at $t_{\text{fact}}$ promoting heads, we have L21H7 (Figure 8a), L20H18 (Figure 8b). Again, with L21H7 and L20H18 we see the complement in categories of Organization and non-Organization, respectively. In this case, the words encoded by the layers are far more interpretable and coherent than for the $t_{\text{cofa}}$ promoting

heads. The words encoded by L20H18 very clearly relate to the topics Location and Language, as expected. The words encoded by L21H7 do not directly relate to organizations, but rather to the topics of food, clothing, and hobby. It should be noted, however, that L21H7 is much less contributing to the competition than L20H18.

These results remain consistent with our findings in the previous experiment, and further support that the heads do not always contribute to the competition between $t_{\text{cofa}}$ and $t_{\text{fact}}$ in general, but are dependent on the domain of information required to correctly answer the prompt.

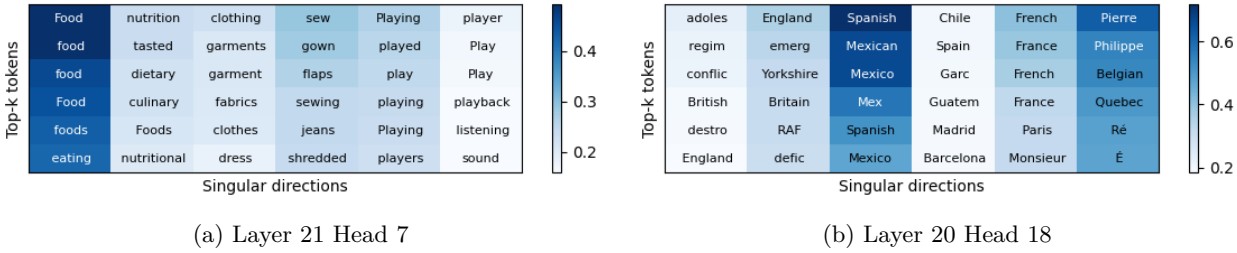

(a) Layer 21 Head 7             (b) Layer 20 Head 18

Figure 8: Heatmap of the top-$k$ tokens of the singular vectors sorted by their singular values.

## 5 Discussion

Generally, our experiment results support the hypothesis 2.1 that when the prompt contains a counterfactual token $t_{\text{cofa}}$, there are attention heads that contribute significantly to the competition between $t_{\text{fact}}$ and $t_{\text{cofa}}$, and moderately manipulating the strengths of these attention heads will affect the proportion of factual and counterfactual outputs accordingly, as the results supporting this hypothesis are reproducible, and the hypothesis can generalize to different premises and sentence structures.

However, this does not necessarily mean that moderately increasing the strengths of these heads will guarantee the generation of more factual responses under all circumstances. When we replace the counterfactual token with the factual token in the original dataset, these heads now suppress the factual token. This suggests that these heads are more likely involved in copy suppression rather than specifically counterfactual suppression, supporting hypothesis 2.2.2 and challenging hypothesis 2.2.1. This also suggests that manipulating the strength of specific attention heads to increase the factual recall rate may not be generally applicable, or needs more improvements and modifications.

Furthermore, we have gained insights into the effect of domain knowledge on how heads contribute to the competition. We have seen that the heads do not uniformly suppress the induction effect across different domains, confirming our hypothesis 2.3.1. But we believe more robust experiments are needed to fully confirm hypothesis 2.3.2. Our results do suggest it is true, but to confirm it, more model sizes within one model family should be tested. We have also investigated further into the information encoded by relevant heads. We find that some heads, most commonly $t_{\text{fact}}$ promoting heads, encode information corresponding to the domain of $t_{\text{fact}}$ and $t_{\text{cofa}}$. Further work could be done by using a counterfact dataset where the domain of the factual and counterfactual tokens differs.

While our research covered multiple scopes and topics, a limitation of our experiments is the fineness of our experiments. More specifically, we did not run the experiments on more models, and we also did not study the attention modification of different heads on different domains. Future work can be in the direction of applying the methods to different models and studying how each attention head interacts with different domains. Considering that the dataset we used is mostly about locations, languages, organizations, and people, constructing new datasets on other domains like STEM and applying similar techniques on them is also a possible direction for future work.

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

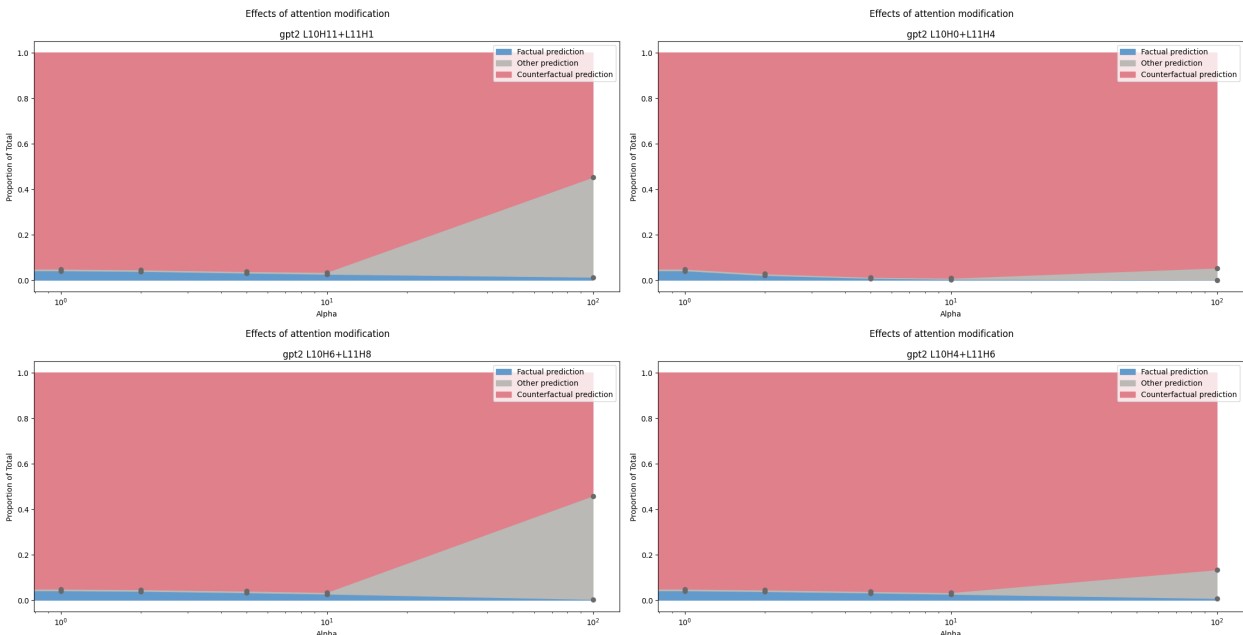

Figure 9: Attention modification on random heads in the same layer as heads of interest.

# A    Computational requirements

We ran our experiments involving GPT-2 on Ubuntu 22.04.2 on a computer with an RTX 4070 12Gb and an Intel(R) Core(TM) i5-9600K CPU @ 3.70GHz. Experiments involving Pythia-6.9B were conducted on a single NVIDIA A100 GPU from the Snellius cluster. In total, we used about 10 hours of GPU time on the RTX 4070, and about 14 hours on the A100.

# B    Additional results

## B.1    Baseline of attention modification

To show that the results from attention modification are results related to the specific heads instead of general results from modifying attention heads, we conducted an experiment by modifying some random attention heads other than the interesting attention heads in the same layer as the heads of interest for GPT-2. The result is shown in Figure 9, under 4 different seeds.

After modification of random attention heads, the model mostly keeps outputting counterfacts under different $\alpha$ values no greater than 10, and above 10, the model may sometimes start to output low-quality predictions that are neither the factual prediction nor the counterfactual prediction. This indicates that, when the $\alpha$ value is moderate, the model's performance would only be significantly changed if the attention heads of interest are modified, while extremely large $\alpha$ values can cause the model to give low-quality predictions due to weights being altered too much.

## B.2    Head selection for comparison in experiment 4.4.1

Figure 10 and 11 were used to identify important heads within and between categories using the setup from 4.4.1. Each plot shows the grid of all heads across all layers, where the color indicates the mean influence of that head on $\Delta_{\mathrm{cofa}}$ within that dataset category. The standard deviations over the categories are shown in Figure 11. This highlights which heads differ the most across categories.

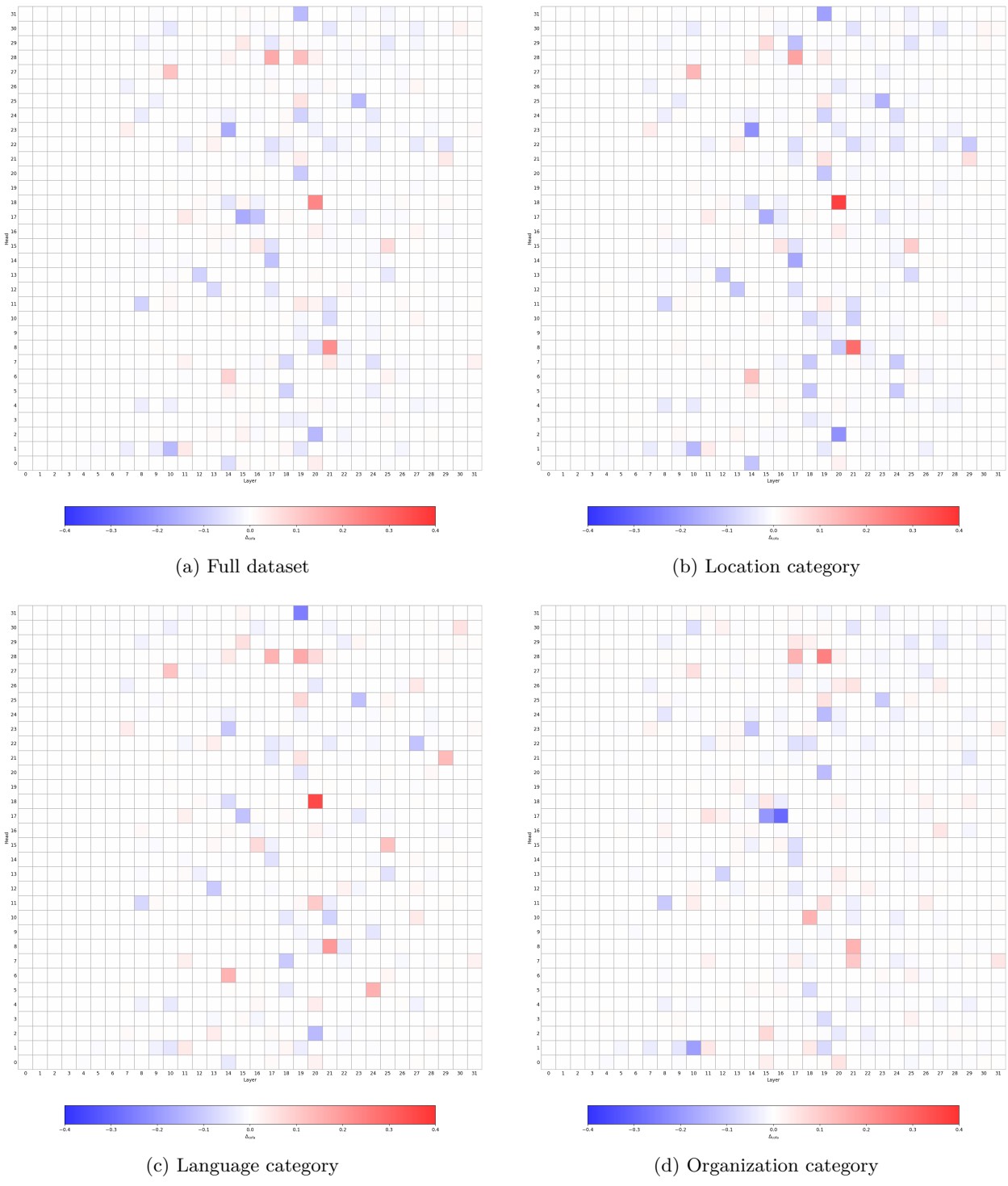

(a) Full dataset

(b) Location category

(c) Language category

(d) Organization category

Figure 10: Mean $\Delta_{\mathrm{cofa}}$ values of each head in Pythia-6.9B for each of the analysed categories

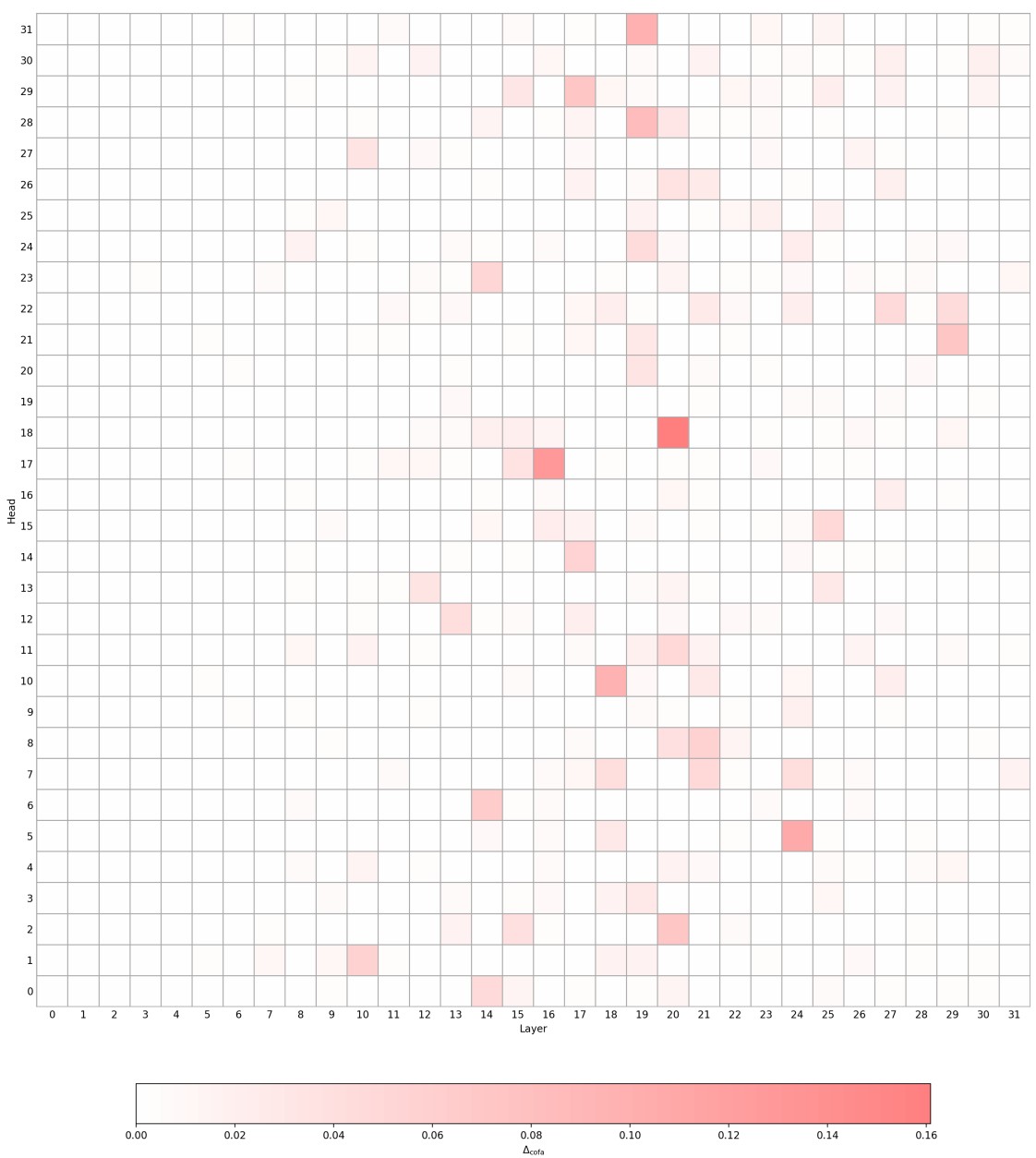

Figure 11: Standard deviation between the mean $\Delta_{\mathrm{cofa}}$ value of the three analyzed categories of the dataset

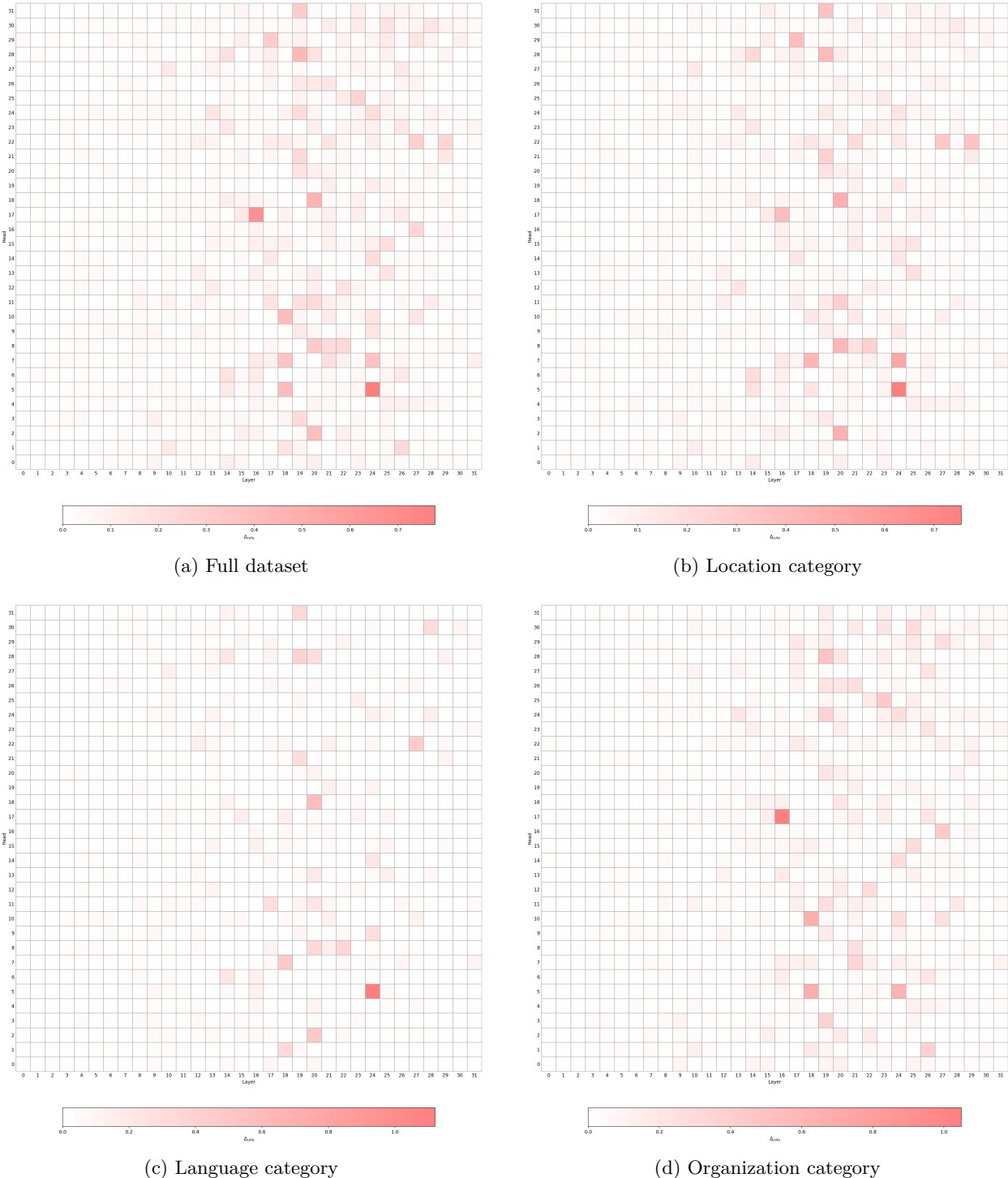

(a) Full dataset

(b) Location category

(c) Language category

(d) Organization category

Figure 12: Standard deviation of $\Delta_{\text{cofa}}$ values of each head in Pythia-6.9B for each the analysed categories

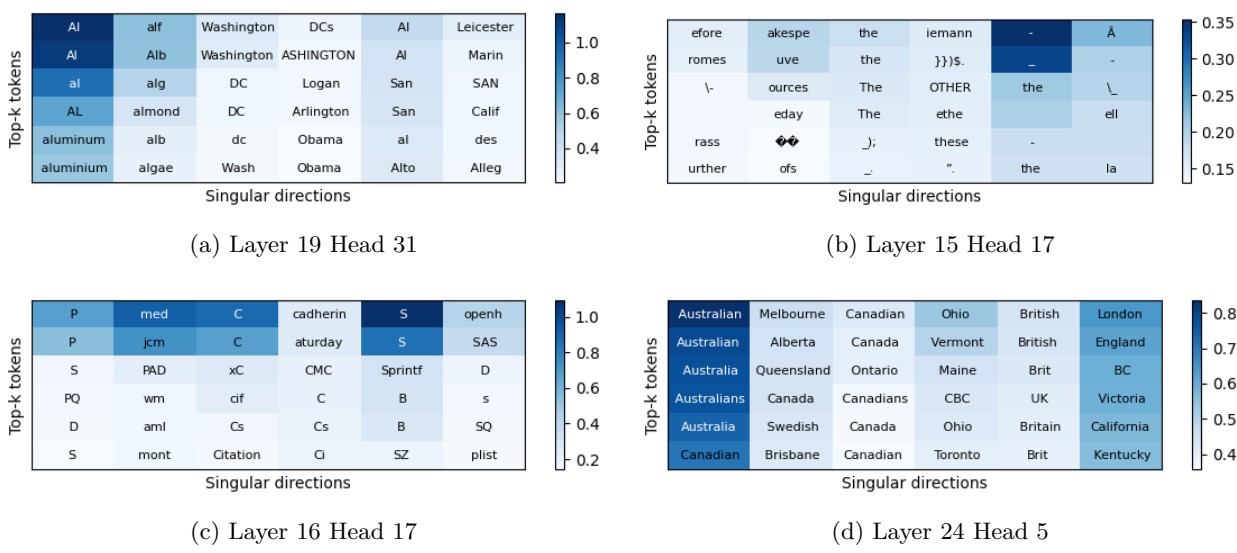

Figure 13: Heatmap of the top-$k$ tokens of the singular vectors sorted by their singular values.

### B.3 Further SVD analysis of relevant heads

Starting with L15H17 (fig 13b), a head that globally promotes $t_{\text{cofa}}$. We do not see many tokens associated with meaning beyond the low-level grammatical structure of language. This is consistent with the finding of Yu, Merullo, and Pavlick [17] that their head contributing most to $t_{\text{cofa}}$ was not related to any specific domain knowledge. Millidge and Black [9] did, however, find that earlier layers more commonly contain such heads encoding punctuation or less interpretable tokens.

Continuing to look at $t_{\text{cofa}}$ promoting heads, L16H17 (Figure 13c) and L19H31 (Figure 13a) seem to behave as each other's complement in categories. Namely, only being effective in the Organization category or only in the non-Organization categories, respectively. Do note that Location and Language are related (e.g., England, English), so this is not too surprising. We see that in this case, there is a minor association with the domain for L19H31, containing some singular directions with geography information, particularly of the United States, but also some with organic and non-organic materials, such as 'aluminum', 'algae', and 'almond'. These do not have a trivial association with the topic of Organizations either however. L16H17 is less interpretable, but notably contains no geographic information.

Our last head, L24H5 (fig 13d), is very interesting, as it significantly supports $t_{\text{cofa}}$ for the Location category but $t_{\text{fact}}$ for the Language category, and has little bias for the Organization category. We see that the words are very interpretable and geographically related, but curiously, almost all words are specifically related to English-speaking commonwealth nations.

