# OpenReview forum: "Tracing Facts or just Copies? A critical investigation of the Competitions of Mechanisms in Large Language Models"
_TMLR — Accepted by TMLR_

### Review · Reviewer_htkn · 2025-03-21

**Summary Of Contributions:**

The paper analyzes LLMs' ability to recall and generate facts when presented with incorrect information in their context, while also identifying specific mechanisms (e.g., attention heads) involved in this process. The authors build upon experimental approaches and findings from three previous works, conducting further analysis to validate their conclusions and test their sensitivity to different experimental variables such as the phrasing and domain areas of input prompts. They begin by describing their hypotheses, the methods they use for their analysis, as well as the datasets and experimental setups. They then present findings from a series of experiments which test these hypotheses. Their key findings reveal that when GPT-2 and Pythia-6.9B encounter incorrect "facts" in their context, specific attention heads can be scaled to increase the models' probabilities of generating correct information. Additionally, the authors demonstrate that this scaling primarily prevents the models from simply copying their context rather than genuinely "recalling" accurate facts. Finally, they show that attention heads' roles vary significantly across different application domains.

**Audience:**

Yes

**Broader Impact Concerns:**

There are no significant ethical concerns regarding this paper. It focuses on interpretability, which contributes positively to machine learning research.

**Claims And Evidence:**

Yes

**Requested Changes:**

Here, I elaborate more on the weaknesses mentioned above, and I provide actionable points that I believe can improve the paper.
* Citations 16 & 21 which the paper builds upon are published, however, they are cited as arXiv preprints.
* I understand that the terminology in the paper is not entirely a choice of the authors, but something that comes from the previous studies they reproduce. However, I found the terms "factual" and "counterfactual" a little problematic. The use of the word "counterfactual" can be misleading to readers with background in causality. In both computer science and psychology, a counterfactual refers to a state of the world under some (past) intervention that could have happened but didn't. In the context of the paper, a counterfact is just something that is factually incorrect (e.g., the capital of France is London). Correct/incorrect would be a much more appropriate terminology with less room for confusion. I would encourage the authors to at least add a footnote clarifying this point.
* The exact problem that the authors are interested in becomes clear quite late in the paper. In the introduction they write "the adaptation of incontext information, come into direct conflict when counterfactual statements opposing the model’s parametric memory are provided as input". This is quite abstract, and a simple example (e.g., the one at the top of page 5) would be helpful to clarify what the paper is about. Also, at the end of page 1, the authors write "Prior work has investigated the competition between factual and counterfactual information in language models, particularly focusing on how different attention heads influence this competition". It is unclear what competition means in this context. I think the authors should provide more examples and concrete explanations in their introduction.
* Although the distinction between hypothesis 2.2.1 and 2.2.2 becomes clear throughout the paper, when they are first stated, their difference is not obvious. This is especially because the both use the term "copying", which if I understood correctly is more related to the anti-induction effect, that is, more related to 2.2.2. I would suggest rephrasing 2.2.1. Again, if i understood correctly, the way I would phrase it is that the first one is about the attention head suppressing incorrect facts, while the latter is about suppressing copies from the context.
* I found the phrasing of hypotheses 2.3.1 and 2.3.2 quite vague. With the current phrasing, they read more like conclusions that followed the experiments, rather than hypotheses that preceded them.
* I found the description of methodology too brief and not precise enough. For example, in "Logit attribution techniques", it is unclear what the word "position" refers to and what "the grouped results are unembedded" means. Perhaps using some formal notation to be precise would be helpful. Similar for the SVD analysis. The authors write "the singular vectors that are of the embedding dimension can be unembedded and detokenized to show what words are most encoded by the head." This sentence gives a vague idea of what the method does but it is quite far from being a rigorous explanation.
* In section 3.2 the authors write "These are the same models that were used in the original paper". It is unclear which paper they refer to here. The one by Ortu et al.?
* The organization of the experimental results is a bit strange, since the authors perform a series of more or less independent experiments, but they first introduce 3 experimental setups and then they present 3 results sections. Maybe this is just my personal preference, but I think the paper would have been much easier to read if each experimental setup was followed by the corresponding results, so that the reader doesn't have to do a lot of back and forth.
* I didn't understand what the first paragraph of page 7 contributes. It introduces a lot of details regarding the CO2 emissions of the study, and it seems unrelated to the actual story of the paper.
* One finding that I find somewhat surprising and worth commenting on is based on Figure 1. To me, it seems that the attention modification is performing best when $\alpha=5$. For GPT-2, $\alpha=10$ gives a slightly higher probability to the correct prediction, but also there is increased mass on predictions that do not match any of the two options (neither the correct one nor the one in the context), which could imply that the model starts generating nonsensical tokens at this point. However, both in GPT-2 and Pythia-6.9B, for $\alpha=5$, both the in-context (incorrect) token and the correct token have probabilities around 0.5, which doesn't seem to imply that, by scaling the parameter, the model "recalls" the correct fact but rather just becomes uncertain. Maybe it would be useful for the authors to comment on that and see if they can identify cases where the model assigns an actually large (>0.5) probability on the correct prediction while still putting most of its mass only on those two tokens.
* In page 7, the part about "Singular value decompositions of transformer." doesn't show any quantitative or qualitative results, however, the authors have a short commentary on their findings. In combination with the very brief description of the corresponding methodology, this contributes to the lack of clarity in the presentation of the paper.
* In 4.2.1, the authors claim that "the effect of changing the attention heads is lower if the two sentences have the same structure", however, that is not immediately clear, since the scales of the corresponding rows in Table 1 are different even for $\alpha=0$. Perhaps showing the percent change as well would be insightful to support the claims. Also, I am not sure if the paragraph that makes a connection with the input length at the end of that section adds much. It doesn't seem to be supported by any numerical results, so it seems a little bit speculative. Also, it seems that $\alpha=10$ leads to more correct predictions in comparison with $\alpha=5$ whenever the first and second sentence structure match. Do the authors have any intuition on why that happens?
* In the beginning of 4.2.2, the authors write "Our results favor hypothesis 2.2.1 more than 2.2.2, showing that the attention heads supporting $t_{fact}$ are more likely to be related with the overall suppression of the induction effect, rather than the selective suppression of counterfactual statements." However, if I am not mistaken, the hypothesis that is claiming this is 2.2.2. Also, in the conclusions, they do write the opposite, that is, "these heads are more likely involved in copy suppression rather than specifically counterfactual suppression, supporting hypothesis 2.2.2 and challenging hypothesis 2.2.1".
* I found the discussion of 4.2.3 somewhat underwhelming. The authors analyze the role of specific attention heads in predicting the correct answer depending on the domain of the input prompt. However, they seem to use a dataset with 124 entries, which seems quite small, and their findings do not include any information regarding the standard deviation of their estimates, so it is quite hard to read the results and make conclusions. Also, it would be helpful to expand the captions of figures 5 and 6 and point to specific rows and columns  when discussing the result. Currently, it is a bit hard to follow. For example the authors write "The heads most supporting $t_{cofa}$ and $t_{fact}$, are L15H17 and L20H18 respectively". It is not clear where a reader is supposed to look to understand that statement.
* The "SVD analysis of relevant attention heads" in page 11 discusses results in figures that are placed in the appendix, which breaks the readability of the paper. I would encourage the authors to either bring those figures to the main body or move that discussion to the appendix and use more space to clarify all the other aspects of their remaining experiments in the main body.
* The paper has several typos, and I think the authors should spell-check it again. Some examples:
    * page 1: "these complexe language model" -> "these complex language model"
    * page 2: "We focus on the works from" -> "we focus on the works from"
    * page 2: "ased on the combined finding" -> "Based on the combined finding
    * page 3: "The anti-induciton effect" -> "The anti-induction effect"
    * pages 3 & 4: "Ortu et al. have provided" should include a citation number
    * page 4: "increasing the over strength" -> "increasing the overall strength"
    * page 6: "To test hypothesis 2.3.1 We wish to see" -> "To test hypothesis 2.3.1, we wish to see"
    * page 7: "align with the the original results" -> "align with the original results"
    * page 12: "mostly about locasions, languages" -> "mostly about locations, languages"

**Strengths And Weaknesses:**

**Strengths:**
1. The paper falls within mechanistic interpretability, a very active area with significant research interest that could appeal to multiple members of the TMLR community. Although largely a reproducibility study, it provides valuable validation (and critique) of previous findings while testing their sensitivity to the experimental setups used in the original studies.
1. The authors' experiments are well-designed and evaluate multiple aspects of the role of attention heads in recalling factual information. I particularly appreciated the experiment examining the "Effect of changing counterfacts to facts in the prompt", which provides insightful evidence that the attention heads under study primarily prevent copying from context rather than directly retrieving facts—an aspect not analyzed in depth by previous studies.

**Weaknesses:**
1. There is limited novelty in terms of technical methodology, which relies primarily on approaches established in prior work. While the paper provides valuable experimental evidence, it introduces little at a methodological level.
1. Parts of the presentation could be further improved for clarity, as some methodologies are described vaguely and certain conclusions are not immediately evident from the presented experimental results.
1. The analysis focuses on only two models (GPT-2 and Pythia-6.9B), leaving unclear whether the findings would generalize to other models/architectures.

---

> ### Author Response · Authors · 2025-05-07
> **Response to Reviewer htkn (1/2)**
>
> Thank you for your elaborate feedback!
>
> Regarding some weaknesses.
>
> We do acknowledge the value of evaluating our approach on various more modern models to demonstrate real-world applicability. Our current submission focuses on establishing fundamental properties using smaller models, as computational and time constraints prevented us from conducting experiments with many model architectures during the review period.
>
> As for the lack of novel methods. This paper is at its core a reproducibility paper, and we found this a particularly interesting avenue for research in part because we believed that the methods already introduced in previous papers had potential for providing further insights.
>
> Our response to the requested changes will appear in the next comment as it did not fit within the character limit.

---

> > ### Author Response · Authors · 2025-05-07
> > **Response to Reviewer htkn (2/2)**
> >
> > Here we list our responses to your requested changes:
> >
> > Point 1, 7, 13, 16: Miscellaneous errors and typos. These have been corrected now, thank you.
> >
> > Point 2:
> > > I understand that the terminology in...
> >
> > We acknowledge the terminological vagueness in our paper. This stems from our reframing of the experiments while using the CounterFact dataset, which inherently presents "counterfactual" statements designed to challenge models' understanding. While Ortu et. al. treat the counterfactual answer as the goal and the factual answer as a mistake, our work shifted focus to the competition between in-context and in-memory answers more generally.  Using the terminology correct/incorrect may be a problem since this very much clashes with the research of Ortu et. al. that we are reproducing, which sees the counterfactual as "correct". So to address your concern, we will add a footnote clarifying our terminology choices and explaining how they relate to the original dataset we utilized.
> >
> > Point 3:
> > > The exact problem that the...
> >
> > We will bring some more attention to this in the introduction. We also plan to add a simple figure 1 to the final paper to make the context of the introduction and hypotheses more clear, including at least the example you mentioned on page 5.
> >
> > Point 4, 5, 6:
> > > Although the distinction between...
> > > I found the phrasing...
> > > I found the description...
> >
> > We will work on clarifying these parts of the paper. Thank you for pointing them out.
> >
> > Point 8 and 9:
> > > organization of the experimental results is a bit strange...
> > > I didn't understand what the first paragraph of page 7 contributes...
> >
> > Although technically preference, we do think that your suggested structure is more practical for most readers. We have reordered those sections. We also agree on your take on the CO2 emission paragraph, so we have removed it, and we have moved the computational cost section to the appendix, which we reference in the experimental setup.
> >
> > Point 10:
> > > One finding that I find somewhat...
> >
> > We think that if the model becomes 'uncertain', we would observe more unrelated predictions instead of correct or in-context predictions. The number 0.5 may just be a coincidence, as it is the highest that correct token probabilities get before the other answers begin to appear. That is, it is the trade-off between 'having meaningful perturbation' and 'disrupting activations so much as to negatively affect the model' on the models we study.
> >
> > Point 11:
> > > In page 7, the part...
> >
> > This part was intending to present that we were able to reproduce the method described by that paper, but indeed this part doesn't actually show any presentable results. To mention that we could reproduce it also is fairly trivial as a few pages later we show results using that technique. We will remove this part.
> >
> > Point 12:
> > > In 4.2.1, the authors claim...
> >
> > To show the effect of changing the attention heads more clearly, we will add the percentage change to our results. For the paragraph mentioning the input length, it was originally for the case where the second sentence is 'normal'. However, after investigation, we agree that the argument lacks strong numerical evidence, and we have decided to remove that paragraph. Instead, we will discuss our theory on why $\alpha=10$ is better when the first and second sentence structures match, while $\alpha=5$ is better when they don't match. Our intuition on why this happens is that, when the two sentences have the same structure, the pattern that induces the induction effect is more prominent, hence it requires a stronger 'anti-induction' effect to suppress the induction effect.
> >
> >
> > Point 14:
> > > I found the discussion of 4.2.3 somewhat...
> >
> > Unfortunately with the distribution of topics within the dataset, when taking the intersection of one category of subject with another category of answer, the number of remaining entries are small. Do note however that this is only for our results comparing the entries filtered on both subject and answer categories, and that is also why we mentioned the low sample size. We will elaborate on this, since it is admittedly left poorly explained, and for all heads we will include the per-category standard deviations in the appendix. The statement "The heads most supporting $t_{cofa}$ and $t_{fact}$, are L15H17 and L20H18 respectively." was referring to the findings of Ortu et. al.. We should have cited there, which we have fixed now. Additionally, we will gladly add to the captions to help point out the observations.
> >
> > Point 15:
> > > The "SVD analysis of...
> >
> > We will move part of those texts to the appendix and add some clarifications in the body.

---

### Review · Reviewer_pXAs · 2025-04-02

**Summary Of Contributions:**

This paper reproduces previous results on mechanisms of attention heads on whether the LLM outputs factual or counter factual information when contextual informaiton is at conflict with the real information, specifically
- The existence of attention heads that influence the encourage factual output (over contextual counter-factual info)
- How positive scaling of target attention heads influence the model factual/counterfactual competition

With the following novel perspectives:
- Whether the attention heads suppress more of counterfactual "token copying" (H2.2.1) or "format matching" ("induction", H2.2.2)
- Whether certain "self-hinting" subject (e.g. Singapore's capital is Singapore) influence this competition
- Whether the attention heads in questions are topic specific (H2.3.1)
- Whether H2.3.1 is sensitive to model sizes (H2.3.2)

**Audience:**

Yes

**Claims And Evidence:**

Yes

**Requested Changes:**

In addition to above:
- Metrics in this work is not consistent. This paper used %$t_{fact}$ and $\Delta_{cofa}$ interchangably. %$t_{cofa}$ should also be presented, or use $\Delta_{cofa}$ consistenly, or explain how they relate.
- Questions: Some previous work you cited also experimented with negative $\alpha$ and had success. Is there a reason you do not experiment with them to better support the attention heads' effects?
- H2.3.2: either have more experiments of models of different sizes in the same model family, or remove any conclusions about it.
- Missing details:
    - Figures 2 and 3 lack descriptions on whether all heads were scaled by the same $\alpha$ or not.
    - Appenfix A.1 paragraph is not complete. The setup is also missing (e.g. what does each entry describe) and not understandable for readers who did not read previous works. (Also Fig 8 fonts too small) This setup could also better be described in S3.4.3
- Typos:
    - `the the` appeared twice.
    - `The anti-induciton effect` -> `The anti-induction effect`
    - `we have reason to believe that they used a different designation than ortu` please cite properly

**Strengths And Weaknesses:**

Strength:
- Good reproduction works highlighting some mismatches
- Novel analysis with fine-grained perspective on self-hinting and topic specialization

Weakness:
- The setup is not fully documented. See requested changes
- Some hypotheses miss more convincing evidence:
    - H2.1
        - "a small number of attention heads contribute most significantly to the competition" is used as a given without experimental evidence. While this probably is true base on Fig 8 estimates, further analsyis is needed. Providing the actual number would help.
        - Discussion to support this hypothesis is minimal. In 4.2.1, no data or figures (e.g. Table 1 should) are cited to prove the boosting effect of factual token logits.
    - H2.2.1 and H2.2.2: promising, but lacks baseline values. It is not very clear whether the output of other tokens (the grey area in the graphs) are a result of "general anti-induction effects", or disruptive scaling of activations. For example, the alpha numbers chosen (2 ,..., 10), while followed previous works, are still not well motivated. Intuitively, you can destroy utility of a model (mabybe just output only "the" or "also") with enough scaling of a single parameter, but we do not know if 10 is close to that disruption or not for the two models. Absolute values here are not interpretable.
    - H2.3.2: more experiments of models of different sizes in the same families must be done to make claim about model sizes.

---

> ### Author Response · Authors · 2025-05-06
> **Response to Reviewer pXAs**
>
> Thank you for your review!
>
> Regarding the metrics, percentages and $\Delta_{cofa}$ serve different purposes. We do not use the notation \%$t_{fact}$ explicitly, but when we use percentages we refer to the percent of output tokens that correspond to $t_{cofa}$ or $t_{fact}$. $\Delta_{cofa}$ is a metric that compares how much a specific head changes the logit values of $t_{cofa}$ and of $t_{fact}$. The former is relevant when showing the effect on the final outputs of the model. The latter is useful to show the relative impact of each head on only $t_{cofa}$ and $t_{fact}$, but it does not take into account the logits of all other tokens. We will add an additional clarification to reduce the chance of misinterpretation in our final version.
>
> We decided against experimenting with negative alpha values, because we were mostly focused on reproducing the paper by Ortu et. al. which do not use negative alpha values. It is true that they did not ground their choice for those, but indeed it had already been shown in Yu et. al. figure 5, 26, and 27, how negative alpha values affect the competition. Negative alpha values in those experiments were mainly shown to effectively boost in-context answers. For the direction we decided to expand in, these measurements did not seem necessary.
>
> We appreciate your insightful observation that our analysis could better differentiate whether the output patterns of other tokens result from anti-induction or disruptive scaling effects. We would need to compare to the typical behavior of ablating any "non-anti-induction" head. To achieve this we will take the average results over a number of random heads that is sufficient yet still feasible in our remaining time and use that as a baseline.
>
> Regarding H2.3.2. That is a great point that we had indeed overlooked. We will make a short analysis on Pythia 1.4 as well so that we can compare within model families.
>
> For figure 2 and 3, all heads we identified in previous passages were scaled with the same alpha value. We will add this to the caption.
>
> Also, thank you for bringing our attention to those typos, they have now been fixed.

---

### Review · Reviewer_iBL3 · 2025-04-23

**Summary Of Contributions:**

This paper presents a reproducibility check of a few recent works in the field of mechanistic interpretability. More specifically, this reproducibility study focuses on three papers studying mechanisms for factual recall, revealing how LLMs dynamically balance pretrained knowledge with contextual knowledge. The contributions are three-fold:
- This paper provides an open-source implementation that replicates the key experiments from the three target papers, which significantly facilitate transparency and reproducibility. This paper and code implementation can serve as a basis for future research in this area.
- The authors compare competing hypotheses regarding the mechanism of factual recalls (hypothesis 2.2.1 vs 2.2.2) with their experiments to establish a clearer consensus in the research community.
- They draw inspiration from one of the reproduced papers and propose two new competing hypotheses (hypothesis 2.3.1 vs 2.3.2) concerning domain specialization, which itself is a new problem in the field. The authors validates the domain specialization hypothesis through their experiments, offering new insight into the encoding of domain-specific factual knowledge in LLMs.

**Audience:**

Yes

**Claims And Evidence:**

Yes

**Requested Changes:**

1. The current experiments are only conducted on GPT-2 and Pythia-6.9B. The factual recall mechanisms discussed may be of great interest to a broader audience, especially those who are working with widely used open-source models (e.g. LLaMA, QWen). If the authors do not consider it beyond the paper's scope, it would be valuable to examine the hypotheses on these popular models and see whether the conclusions are generalizable to different models.
2. The summary of conclusions does not appear until the last section. Reorganizing the paper to bring a summary of the main conclusions earlier in the paper could make it more informative to the readers interested primarily in the conclusions instead of the methodological or technical details.
3. As is mentioned in weakness 1, I would recommend a better justification for the motivation behind selecting this scope of the paper. Providing more discussion on why the three papers are chosen helps strengthen the overall narrative of this paper.

**Strengths And Weaknesses:**

Strengths:
1. The experiments are thorough, and the conclusions are convincingly supported by the experiments. The experiments can sufficiently distinguish between the proposed competing hypotheses.
2. The topic itself could interest a lot of people. Providing explanations for the factuality of current LLMs might be appealing not only to mechanistic interpretability researchers, but also the broader communities interested in trustworthy AI systems.

Weaknesses:
1. The paper has made its scope of reproducibility very clear, but the motivation behind the chosen scope is insufficiently expressed. Despite all three papers examining the factual recall mechanism of transformer-based LLMs, the reason why the specific three works are selected is not adequately justified. The readers could be uncertain whether these works are the most representative and most recent ones in the area. Furthermore, from my perspective, it would be beneficial if a further discussion on how these three studies are positioned within the broad landscape of mechanistic interpretability.
2. Despite systematic and sound experimental designs, the methods that the authors adopt to explain factuality mechanisms are not novel. This paper can benefit from a novel theoretical framing that provides new insights into mechanistic interpretability.

---

> ### Author Response · Authors · 2025-05-07
> **Response to Reviewer iBL3**
>
> Thank you for your review!
>
> We acknowledge the value of evaluating our approach on various more modern models to demonstrate real-world applicability. Our current submission focuses on establishing fundamental properties using smaller models, as computational and time constraints prevented us from conducting experiments with many model architectures during the review period.
>
> Presenting the main conclusions earlier in the paper is a welcome suggestion. We will certainly add a short summary of our findings to our abstract and introduction.
>
> Regarding the reasoning behind choosing these specific papers. For the [MLRC](https://reproml.org), we selected the paper by Ortu et al. as our primary reproduction target. This choice was motivated by the paper's interesting claims and its publication as a full paper at a major conference, meeting the challenge requirements. However, as we delved into our analysis, we discovered several related works that either supported or challenged Ortu et al.'s findings. We felt that conducting a more comprehensive examination by highlighting these most relevant comparative papers would provide greater context and insight into the reproducibility of the original claims. While we considered expanding our scope further to include additional related publications, we ultimately limited our analysis to maintain focus and ensure thorough treatment of the most directly relevant papers. Including more works would have expanded beyond the feasible scope of this challenge submission.
>
> Regarding the lack of novel methods. This paper is at its core a reproducibility paper, and we found this a particularly interesting avenue for research in part because we believed that the methods already introduced in previous papers had potential for providing further insights. We believe that the audience of TMLR will still be interested.

---

> > ### Comment · Reviewer_iBL3 · 2025-05-07
> >
> > Dear Authors,
> > Thank you for your thoughtful rebuttal. It satisfactorily addresses the concerns I raised in my initial review. Your clarifications and responses have resolved my questions.

---

### Decision · Action_Editor_LXNK · 2025-06-09

**Recommendation:** Accept with minor revision

**Additional Comments:**

The paper reproduces three different works in the space of attention head attribution. It aims to test several hypotheses through standardized experiments across these methods, evaluating whether findings from the papers hold when experimental conditions are changed.

**Audience:**

Yes

**Audience Explanation:**

This is an area a lot of people in TMLR's audience care about, and the paper provides more thorough experiments than the works it aims to reproduce. Therefore, I expect this to be of interest.

**Claims And Evidence:**

Yes

**Claims Explanation:**

The paper evaluates the role of attention heads in recalling factual information. They provide thorough, open-source reproductions of several studies in this space. The reviewers describe the experiments as well-designed and thorough. There are concerns from reviewer pXAs that some of the findings are overstated relative to what can be backed by the experiments, but I believe these can be resolved by toning down the language a bit, without the need for additional experiments.

Note to authors: there are a handful of typos I still noticed in the draft (e.g. "srenght" instead of "strength", "ased" instead of "Based", etc.). Please do a careful round of editorial revision before submitting your next version.